# Effects of Maternal Supplementation with Organic Trace Minerals including Zinc, Manganese, Copper, and Cobalt during the Late and Post-Partum Periods on the Health and Immune Status of Japanese Black Calves

**DOI:** 10.3390/ani13233679

**Published:** 2023-11-28

**Authors:** Amany Ramah, Tomohiro Kato, Urara Shinya, Mahmoud Baakhtari, Shoichiro Imatake, Arvendi Rachma Jadi, Masahiro Yasuda

**Affiliations:** 1Graduate School of Medicine and Veterinary Medicine, University of Miyazaki, Miyazaki 889-2192, Japan; amanyra@cc.miyazaki-u.ac.jp (A.R.); gf14003@student.miyazaki-u.ac.jp (S.I.); arvendi_rachma_jadi@med.miyazaki-u.ac.jp (A.R.J.); 2Department of Forensic Medicine and Toxicology, Faculty of Veterinary Medicine, Benha University, Qalyubia 13518, Egypt; 3Kagoshima Agricultural Mutual Aid Association Soo Station, Soo 889-8212, Japan; kato-t@nosai46.jp (T.K.); shinya-u@nosai46.jp (U.S.); 4Laboratory of Veterinary Anatomy, Faculty of Agriculture, University of Miyazaki, Miyazaki 889-2192, Japan; mahmoud.baakhtari@yahoo.com; 5Department of Anatomy, Faculty of Veterinary Medicine, Universitas Gadjah Mada, Yogyakarta 55281, Indonesia

**Keywords:** immunity, Japanese black calf, late gestation, respiratory disease, trace minerals

## Abstract

**Simple Summary:**

Beef calves are born with underdeveloped immune systems and extremely vulnerable to diseases. Therefore, calf health is one of the most significant animal health issues facing the livestock industry. Maternal nutrition during pre- and post-partum with essential nutrients plays a significant role in offspring’s physiological functions. Thus, this study attempted to evaluate the effects of supplementation of the maternal diet with organic trace minerals on the health and immune status of beef calves. This study indicated that maternal supplementation with trace minerals containing zinc, manganese, copper, and cobalt is a promising strategy for preventing infections and improving calves’ immunity.

**Abstract:**

In this study, we evaluated the effects of supplementation of the maternal diet with organic trace minerals including Zn (zinc), Mn (manganese), Cu (copper), and Co (cobalt) on the health and immune status of beef calves. We examined 19 pregnant cows, which were divided into a group of 9 cows fed a basal diet (control) and 10 cows fed a diet with organic trace minerals (treated). Cows were fed for a period of 45 days before the predicted calving date until 45 days after calving. The number of treatments needed for respiratory and digestive diseases within 14 days of birth was significantly lower in the treated group (*p* < 0.05) than the control group. In addition, the concentration of serum zinc in the treated group on day 1 was significantly higher (*p* < 0.05) than that in the control group. The numbers of CD4^+^ and CD8^+^ cells in the treated group on days 30 and 60 were significantly increased (*p* < 0.01) compared with those in the control group, as was the number of γδ T cells on days 1 and 30 (*p* < 0.05). The number of IgM^+^ cells in the treated group on days 30 and 60 was significantly increased (*p* < 0.01) compared with that in the control group, as was the number of MHC class II^+^ cells on day 60 (*p* < 0.01). The number of NK cells in the treated group on day 60 was also significantly increased (*p* < 0.05) compared with that in the control group. The expression levels of mRNAs encoding interlukin-2 (IL-2), interlukin-4 (IL-4), interlukin-12 (IL-12), and interferon-γ (IFN-γ) in the treated group were significantly higher than those in the control group (*p* < 0.05) on days 1 and 60. The results indicate that maternal supplementation with trace minerals is a promising approach for producing highly disease-resistant calves and enhancing calf immunity.

## 1. Introduction

In the beef industry, nutritional status has significant implications for production systems. Therefore, maternal nutrition has a significant impact on neonatal calf development, including differentiation, organogenesis, and vascularization immediately post-calving [1,2]. At birth, calves are exposed to stressors such as respiratory and digestive diseases due to a variety of factors, including (i) maternal sufficiency of nutrients, such as trace minerals, or deficiency of these minerals, (ii) lack of roughage for mother cows, (iii) frequent feeding content changes, and (iv) unfavorable barn conditions, all of which may affect developmental processes and postnatal function [3,4,5]. In such cases, economic losses are likely to be significant due to increased mortality. Therefore, manipulation of maternal nutrition to improve health during the neonatal stage and strengthen immunity after birth is important [6].

Trace minerals such as zinc (Zn), copper (Cu), manganese (Mn), and cobalt (Co) play important roles in beef cow health and production during late gestation, and deficiencies in these minerals during fetal life can cause negative effects on the growth and immunological development of neonatal calves [7,8,9,10]. These trace minerals are essential for protein synthesis, lipid metabolism, and bone formation during neonatal calf development [3,11], and they are also required for proper development of the nervous, reproductive, and immune systems [11,12]. Cattle are supplemented with minerals in the form of organic forms of trace minerals, including minerals complexed with amino acids, which exhibit greater bioavailability and absorption than inorganic forms, thereby reducing the risk of mineral antagonistic interactions [11,13,14,15].

Previous reports have revealed that calves born from cows supplemented with organic complexes of Zn, Cu, Co, and Mn have higher liver concentrations of Co, Cu, and Zn at birth. Additionally, an increased body weight of calves at weaning and a lower incidence of bovine respiratory diseases have been reported compared with calves from both non-supplemented and inorganically supplemented cows [16]. Another study reported similar results for the offspring of sows supplemented with organic Cu, Mn, and Zn and revealed that the concentrations of these elements were higher in fetal tissues; fetal loss was also reduced by day 30 of gestation [11]. The authors also reported that these findings were associated with enhanced transfer of Zn and Cu from maternal to fetal tissues, which had long-lasting programming effects on the productivity and health of offspring [17].

Few studies have examined how organic trace minerals affect disease resistance and immune status in calves, however. Based on previous studies, we hypothesized that providing beef cows with organic trace minerals during the last trimester of pregnancy would be an additional way to enhance postnatal progeny productivity. The objective of this study, therefore, was to determine the effects of maternal supplementation with organic trace minerals during the late and post-partum periods on the health and immune status of Japanese black (JB) calves.

## 2. Materials and Methods

All procedures used in this study were conducted according to protocols approved by the Animal Care and Use Committee of the University of Miyazaki (approval no.: 2019-001-02).

### 2.1. Cow Management and Dietary Treatments

A total of 19 healthy JB pregnant beef cows, with a body weight around 500 kg, and a body condition score ranging from 3 to 5 were kept at a large-scale breeding farm in Kagoshima Prefecture. The cows were impregnated via artificial insemination (AI). All diets were isocaloric and isonitrogenous and formulated to meet requirements for energy, protein, macrominerals, selenium (Se), iodine (I), and vitamins for pregnant cows during the last trimester of gestation [18]. The management of caws and calves and diet composition (Appendix A) is shown in the Appendix A.

Before the initiation of the study, the cows were divided into two groups, a control group (age 3.8 ± 1.1 years old, *n* = 9) and a treated group (age 3.7 ± 1.1 years old, *n* = 10). The control group’s diet was free of supplemental trace mineral preparations of Zn, Mn, Cu, and Co, whereas the diet of the treated group contained 36 mg/kg Zn provided as an amino acid chelate Zn, 20 mg/kg Mn provided as an amino acid chelate Mn, 12.5 mg/kg Cu provided as an amino acid chelate Cu, and 2.5 mg/kg Co provided as a Co sulfate (10 g/day of Availa 4, Zinpro Corp., Eden Prairie, MN, USA). The supplement was mixed with the feed, and cows were fed individually (twice/day) for a period 45 days before the predicted calving date until 45 days after calving (Figure 1). The chest circumference of calves was measured on days 1, 30, and 60. The incidence and number of treatments for respiratory and gastrointestinal diseases in calves up to day 30 were also evaluated.

### 2.2. Sample Collection

After the calving period, blood samples were obtained from neonatal calves on days 1, 30, and 60 via jugular vein puncture, and the blood was transferred into sterile ethylenediamine tetraacetic acid (EDTA)-containing tubes. The samples were analyzed for lymphocyte subsets immediately after transfer. Blood samples were collected and transferred into tubes, and serum was then isolated for the measurement of Zn concentrations using metal assay zinc determination kit (Metallogenics Co., Ltd., Tiba, Japan).

### 2.3. Flowcytometry

#### 2.3.1. Cell Preparation

Peripheral blood mononuclear cells (PBMCs) were isolated from whole blood using standard Ficoll-Paque (GE Healthcare UK, Little Chalfont, Buckinghamshire, UK) density gradient centrifugation. The layer of PBMCs was harvested and the cell pellet was placed in 10 mL of RBC lysis buffer according to our previous report [19]; the PBMCs were pelleted via centrifugation at 300× *g* at room temperature for 10 min after washing with 10 mL. The supernatant was discarded, and cells were suspended in phosphate buffer saline (PBS) supplemented with 0.5% bovine serum albumin (Nacalai Tesque, Inc., Kyoto, Japan) and 0.05% sodium azide (Fujifilm Wako Pure Chemical Co., Osaka, Japan) (BSA-PBS).

#### 2.3.2. Cell Sorting

The viable cells, whose concentration ranged from 1 × 10^5^ to 1 × 10^6^ cells, were incubated for 60 min on ice with fluorophore-labeled monoclonal antibodies (mAbs), and then BSA-PBS was added three times to the stained cells, and the cells were centrifuged for 3 min and resuspend in an appropriate volume of BSA-PBS containing propidium iodide (1 µg/mL, Sigma-Aldrich, St. Louis, MO, USA). The FACS Canto^TM^ II system (Becton Dickinson, Franklin Lakes, NJ, USA) was used to analyze relative immunofluorescence intensities using flow cytometry. Anti-CD4 (1:200 dilution, ILA11A, Monoclonal Antibody Center at Washington State University, Pullman, WA, USA), anti-CD8 (1:200 dilution, CC63, Monoclonal Antibody Center at Washington State University), anti–γδ TCR (1:200 dilution, GB21A, Monoclonal Antibody Center at Washington State University, Pullman, WA, USA), anti–MHC class II (1:200 dilution, TH14B, Monoclonal Antibody Center at Washington State University, Pullman, WA, USA), and anti-IgM (1:100 dilution, BIG73A, Monoclonal Antibody Center at Washington State University, Pullman, WA, USA) were used. The FITC labeling kit for NH_2_, HiLyte^TM^ Fluor 555 (F555) labeling kit for NH_2_, and HiLyte^TM^ Fluor 647 (F647) labeling kit for NH_2_ (Dojindo Laboratories, Kumamoto, Japan) were used according to the manufacturer’s instructions. Absolute white blood cell (WBC) numbers were counted using a pocH-100iV Diff (Sysmex, Hyogo, Japan). The percentage of lymphocytes in WBCs was measured via Giemsa staining of blood smears. The following formula was used to calculate the number of cells in each subset, according to our previous work [20]. Cell density (cells/µL) = relative population of cells estimated according to multicolor fluorescence-activated cell sorting (FACS) × proportion of lymphocytes in WBCs determined from smears × number of WBCs.

### 2.4. Total RNA Extraction and Real-Time PCR Analysis of Cytokines

Total RNA was isolated from a frozen mononuclear pellet using an RNeasy Mini Kit (Qiagen, Valencia, CA, USA) following the manufacturer’s protocol for the isolation of total RNA from animal cells. Real-time RT-PCR was carried out using a one-step TB Green PrimeScript PLUS RT-PCR kit (Takara Bio., Tokyo, Japan) according to the manufacturer’s protocol. The following conditions were used: reverse transcription at 42 °C for 5 min and initial PCR activation at 95 °C for 10 s, followed by 40 cycles at 95 °C for 5 s, 57 °C for 30 s, and 70 °C for 30 s; the results were confirmed via melting curve. Quantitative real-time PCR was established to analyze the expression levels of glyceraldehyde-3-phosphate dehydrogenase (GAPDH), interlukin-1β(IL-1β), interlukin-2 (IL-2), interlukin-4 (IL-4), interlukin-6 (IL-6), interlukin-10 (IL-10), interlukin-12 (IL-12), and interferon-γ (IFN-γ) using the QuantStudio™ Real-Time PCR system (Applied Biosystems, Carlsbad, CA, USA). The primer sequence was designed using Oligo 7 software (Molecular Biology Insights, Colorado Springs, CO, USA) and is shown in Table 1. Glyceraldehyde phosphate dehydrogenase (GAPDH) was used to standardize the mRNA expression level of target genes, which was calculated based on the comparative Ct method (2^−ΔΔCt^ method). QuantStudio™ software (Thermo Fisher Scientific, Waltham, MA, USA) was used to analyze the data [21].

### 2.5. Statistical Analysis

Data were analyzed using R statistical software, version 3.5.1 (R Core Team, Vienna, Austria). The Mann–Whitney U test was used to determine significant differences between the experimental groups. A nonparametric test (Log-rank test) was used to determine difference in the incidence rate between the experimental groups. Results are expressed as mean ± SD, with *p* values < 0.05 considered statistically significant.

## 3. Results

### 3.1. Chest Circumference, Respiratory and Digestive Disease Incidence, and Treatment

The results of chest circumference measurements during the study period are shown in Figure 2A. The chest circumference of the treated group on days 1, 30, and 60 was slightly increased compared with the control group, but no significant differences were observed between groups.

The effects of maternal supplementation with trace minerals on the incidence of respiratory and digestive disease in calves, as well as the treatments, are shown in Figure 2B,C. The incidence rate up to 30 days did not differ between groups but was 30% in the treated group and 62.5% in the control group (Figure 2B). However, the number of treatments required within 14 days was significantly lower in the treated group (*p* < 0.05) compared with the control group (Figure 2C). There were no differences within 30 days (*p* = 0.106).

### 3.2. Zn Concentration

As shown in Figure 3, compared with the control group, on day 1, calves in the treated group showed an increase in serum Zn concentration (*p* < 0.05). However, there were no significant differences between the groups on day 30.

### 3.3. Lymphocyte Count and Percentages of Lymphocytes and Monocytes

As shown in Figure 4A,B, respectively, lymphocyte and monocyte percentages were significantly higher in the treated group at all stages compared with the control group. Moreover, the lymphocyte count was significantly higher in the treated group on days 30 and 60 than in the control group, as shown in Figure 4C.

### 3.4. Lymphocyte Subset Analysis

T-lymphocyte subset analysis showed a significantly higher number of CD4^+^ cells and CD8^+^ cells (*p* < 0.05 and *p* < 0.01) in the treated group compared with the control group on days 30 and 60, as shown in Figure 5A,B. Additionally, the number of γδ T cells was significantly higher on days 1 and 30 in the treated group compared with the control group (*p* < 0.05) (Figure 5C).

The results of lymphocyte subset analysis show that the number of major histocompatibility complex (MHC) class II^+^ cells was significantly higher in the treated group than in the control group on days 1 and 60 (Figure 5D; *p* < 0.05 and *p* < 0.01, respectively). The number of immunoglobulin M (IgM)^+^ cells was significantly higher in the treated group on days 30 and 60 compared with the control group (Figure 5E; *p* < 0.01). In addition, as shown in Figure 5F, the number of natural killer (NK) cells was significantly higher in the treated group compared with the control group on day 60 (*p* < 0.05).

### 3.5. Cytokine-Encoding mRNA Expression

The expression of cytokine-encoding mRNAs in 1-day-old and 60-day-old calves in the treated and control groups is shown in Figure 6A,B, respectively. The levels of mRNAs encoding IL-2, IL-4, IL-6, IL-12, and IFN-γ were significantly higher in the treated group than the control group on day 1 (*p* < 0.05). In addition, the levels of mRNAs encoding IL-2, IL-4, IL-12, and IFN-γ were significantly higher in the treated group than the control group on day 60 (*p* < 0.05).

## 4. Discussion

Ensuring the health and safeguarding the welfare of newborn calves is one of the most significant challenges facing the livestock industry. Additionally, because the immune system of neonatal calves is still developing, they are more susceptible to infectious pathogens. Therefore, calves rely on the effective passive transfer of maternal Ig from the colostrum after birth [22,23,24,25,26].

Trace elements, which play significant roles in nutrition and regulate many critical biological processes, are important for optimizing the production of beef cattle [27,28,29,30,31]. Trace mineral deficiencies or impaired placental transfer of these minerals during fetal life can adversely affect not only overall growth but also the immunological and morphological development of a variety of fetal and neonatal tissues [32,33]. Therefore, the present study hypothesized that feeding organic trace minerals to cows during the late and post-gestation periods could reduce the incidence of respiratory and gastrointestinal diseases and improve the immune status of calves.

In the present study, calves did not exhibit a difference in chest circumference or growth rate in the treated group compared with the control group. Our results agree with previous studies reporting that supplementing the diet of late-gestating beef cows with trace minerals, as organic or inorganic sources, does not impact calf body weight at birth [7,10,34]. A similar result was reported in dairy cows [35]. These outcomes may be attributed to supplementing the diet with an organic trace mineral source that may be even more beneficial to offspring development if offered to beef cows over a greater duration of gestation. In addition, Marques et al. revealed that at weaning time, calves born to cows supplemented with organic trace minerals exhibited a weaning body weight more than 20 kg higher than that of control cows [8,16].

The number of treatments for respiratory and gastrointestinal diseases in the treated group was lower than that in the control group in the present study. Previous findings indicate that there is a lower incidence of respiratory diseases in calves born to cows fed an organic-complexed source of trace elements during the feedlot phase [8,36].

As our data revealed, the serum Zn concentration on day 1 was significantly elevated in the treated group compared with the control group. Previous research has demonstrated that Zn plays important roles in offspring health, productivity, and the modulation of immune function [37]. According to Hostetler et al. (2003), fetuses from sows supplemented with organic sources of Cu, Mn, and Zn have higher amounts of these elements in their tissues than fetuses from sows supplied inorganic sources. These findings indicate that the maternal colostrum contains trace minerals that are transferred to neonatal calves on the first day of life [11]. Another study has also indicated that at weaning, steer plasma zinc status was improved when gestating dams received an injectable trace mineral supplement [38].

The ratios of lymphocytes to monocytes at all stages, as well as the number of lymphocytes, were significantly increased on days 30 and 60 in the treated group. In addition, the populations of CD4^+^ cells and CD8^+^ cells were significantly higher on days 30 and 60 in the treated group. Other previous studies reported that trace minerals such as Zn, Mn, Co, and Se are important for optimal immune function [13,33]. Previous studies have also indicated that supplementing maternal nutrition with organic trace elements affects the neonatal innate immunity response at least in part via changes in gene and mRNA expression [35]. Another study reported that Zn is essential for lymphocyte proliferation and differentiation as well as DNA replication via the activity of ribonucleotide reductase [37]. According to these authors, maternal supplementation with organic trace elements can affect the newborn’s immune response.

CD4^+^ T cells differentiate into multiple types of immune effector cells with different functions that enhance and regulate the immune response [39,40]. For example, CD4^+^ cells secrete IFN-γ to activate cellular immune responses and IL-10 to regulate inflammatory responses [41,42]. CD4^+^ cells also induce the development and maintenance of protective effector memory CD8^+^ cell responses during viral infection or immunization [41,43]. In healthy cows, the majority of CD8^+^ T cells are extravascular T cells that preferentially travel to the tissues [44,45]. These cells are hypothesized to be involved in tumor regression by eliminating altered cells after activation by T-helper cells, and they are also thought to play roles in protection and recovery from viral, certain bacterial, and parasite infections [46,47,48].

The number of γδ T cells in the peripheral blood is typically very high in healthy young calves, and this population can constitute up to 60% of circulating T cells [49,50,51,52]. In our study, the number of γδ T cells was significantly higher in the treated group than in the control group on days 1 and 30. Moreover, γδ T cells play an essential role in the initial immunity of calves after infection as well as a critical role in anti-viral immune responses [53]. Several researchers reported that γδ T cells in the peripheral blood play an important role in bridging the innate and adaptive immune responses and that γδ T cells function as the principal immune regulatory subset in the ruminant immune system [52,54]. These data suggest that supplementing maternal nutrition with trace minerals can accelerate the maturation of the adaptive arm of the neonatal immune system in calves.

In this study, the number of MHC class II^+^ cells was significantly higher on day 60 than on day 1. MHC class II is expressed on dendritic cells (DCs), B cells, and macrophages, which are regarded as major antigen-presenting cells [55]. DCs and macrophages also activate and induce the differentiation of T lymphocytes [56]. Interestingly, an age-associated increase in the percentage of MHC class II^+^ cells has been reported [57]. The longitudinal changes in leukocyte subsets seen in calves likely reflect the maturation of the calf’s immune system [58,59,60].

In addition, the percentage of IgM^+^ cells in the treated group was significantly higher on both days 30 and 60 compared with the control group. IgM is an important antibody isotype produced during primary immune responses [61,62]. We speculate that the increase in the number of IgM^+^ cells in the treated group indicates the stimulation of humoral immunity.

The number of NK cells was also significantly higher on day 60 in the treated group. Previous studies have indicated that NK cells play an important role in inducing early immunity in calves vaccinated against viral infections [63,64]. Our data suggest that the increased number of NK cells in the treated group is indicative of increased protection against viral infection.

The levels of mRNAs encoding IL-2, IL-4, IL-6, IL-12, and IFN-γ were significantly higher in the treated group than the control group on day 1 in the present study. The colostrum, which contains cytokines, lymphocytes, and other elements, stimulates infant immunity [65,66]. Cytokines are soluble mediators involved in the regulation of the immune response [67]. As a consequence, the maternal colostrum plays an important role in the regulation of both cellular and innate immunity.

On day 60 in the present study, the treated group exhibited a significant increase in the expression of mRNAs encoding IL-2, IL-4, IL-12, and IFN-γ relative to the control group. According to previous studies, IL-2 regulates the growth, homeostasis, and function of distinct T-cell subsets, promotes the early proliferation of naïve T cells in the thymus, and mediates Treg maturation and effector T-cell activity [68,69,70,71]. Moreover, IFN-γ produced by T-helper 1 (Th1) cells is responsible for the regulation of cell-mediated immune responses involved in the eradication of intracellular and viral pathogens [72]. IL-4 is produced by Th2 cells and induces the differentiation of B cells and plays an essential role in IgM to IgG class switching. IL-4 is also important for the defense response to parasitic infections and the inactivation of toxins [73,74,75]. IL-12, which is produced by macrophages and DCs, stimulates the production of IFN-γ, promotes the development of Th1 cells, and functions in bridging innate and adaptive immunity [76,77,78]. Overall, the available data suggest that maternal nutrition plays an important role in immune function and health in neonatal calves.

## 5. Conclusions

The present data underscore the importance of feeding organic trace minerals including Zn, Mn, Cu, and Co to cows during the pre-and post-partum period, as it contributes to a reduction in the incidence of respiratory and gastrointestinal diseases in calves. In addition, the enhancement of acquired immune function in the calves of cows supplemented with organic trace minerals is due to increases in the numbers of various lymphocyte subsets and the upregulated expression of certain cytokines. Collectively, these findings indicate that supplementation of the maternal diet with trace minerals including Zn, Mn, Cu, and Co could be useful, as it contributes to the production of highly disease-resistant JB calves.

## Figures and Tables

**Figure 1 animals-13-03679-f001:**
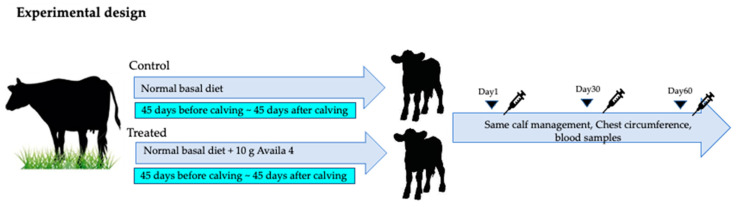
Experimental design. The cows were divided into a control group fed a normal basal diet and a treated group fed a 10 g Availa 4 diet for 45 days pre- and post-partum (twice/day). The chest circumference of calves was measured, and blood was collected on days 1, 30, and 60.

**Figure 2 animals-13-03679-f002:**
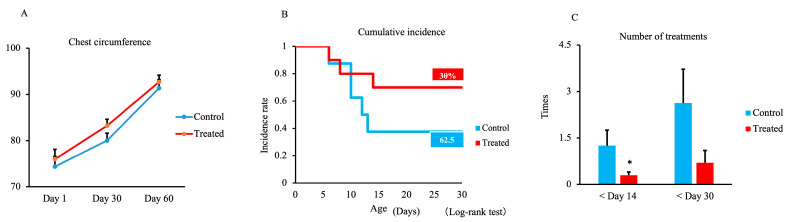
Effect of maternal supplementation with organic trace minerals on chest circumference, cumulative disease incidence, and number of treatments needed for respiratory and gastrointestinal diseases in Japanese black calves. (**A**) Chest circumference was measured on days 1, 30, and 60. (**B**) Cumulative incidence within 30 days and (**C**) the number of treatments within 14 and 30 days. The control group was fed a normal basal diet, and the treated group was fed a 10 g Availa 4 diet for 45 days pre- and post-partum. Values represent the mean ± SD. * *p* < 0.05 compared with control.

**Figure 3 animals-13-03679-f003:**
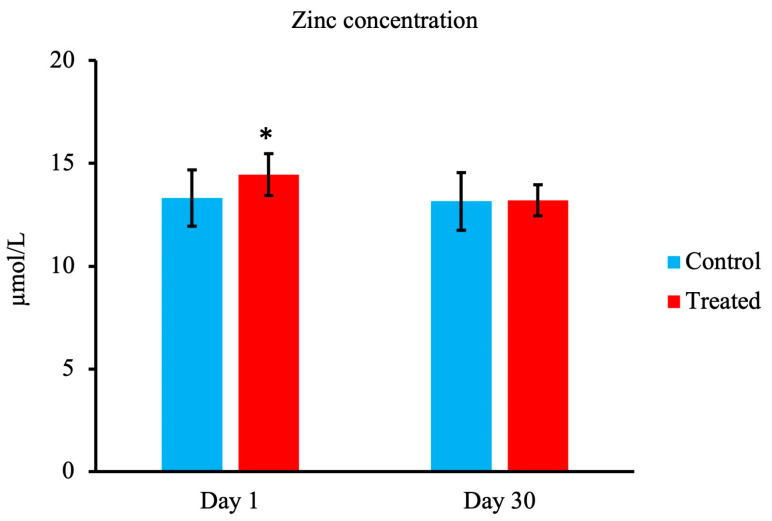
Effect of maternal supply of organic trace minerals on serum Zn concentration of Japanese black calves. The serum Zn concentration was measured on days 1 and 30. The control group was fed a normal basal diet, and the treated group was fed a 10 g Availa 4 diet for 45 days pre- and post-partum. Values represent the mean ± SD. * *p* < 0.05 compared with control.

**Figure 4 animals-13-03679-f004:**
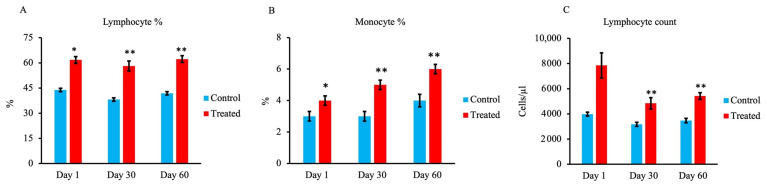
Effect of maternal supply of organic trace minerals on lymphocyte, monocyte, and lymphocyte counts of Japanese black calves. (**A**) Percentage of lymphocytes, (**B**) percentage of monocytes, and (**C**) lymphocyte count were recorded on days 1, 30, and 60. The control group was fed a normal basal diet, and the treated group was fed a 10 g Availa 4 diet for 45 days pre- and post-partum. Values represent the mean ± SD. * *p* < 0.05, ** *p* < 0.01 compared with control.

**Figure 5 animals-13-03679-f005:**
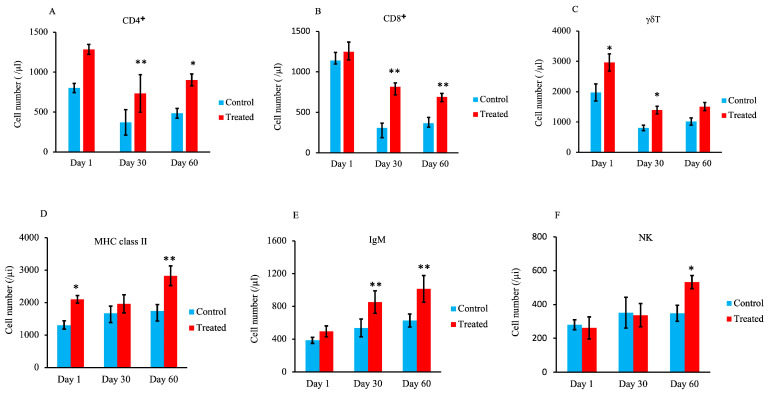
Effect of maternal supply of organic trace minerals on lymphocyte populations in Japanese black calves. Numbers of CD4^+^ cells (**A**), CD8^+^ cells (**B**), and γδ cells (**C**) are shown. In addition, the numbers of MHC class II^+^ cells (**D**), IgM^+^ cells (**E**), and NK cells (**F**) were recorded on days 1, 30, and 60. The control group was fed a normal basal diet, and the treated group was fed a 10 g Availa 4 diet for 45 days pre- and post-partum. Values represent the mean ± SD. * *p* < 0.05, ** *p* < 0.01 compared with control.

**Figure 6 animals-13-03679-f006:**
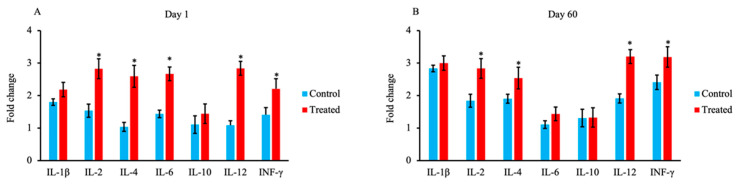
Effect of maternal supply of organic trace minerals on the expression of mRNAs encoding IL-1β, IL-2, IL-4, IL-6, IL-10, IL-12, and IFN-γ in Japanese Black calves. The fold change in gene expression in JB calves was measured on day 1 (**A**) and day 60 (**B**). Expression levels were measured using real-time PCR. The control group was fed a normal basal diet, and the treated group was fed a 10 g Availa 4 diet 45 days pre- and post-partum. Values represent the mean ± SD. * *p* < 0.05 compared with control.

**Table 1 animals-13-03679-t001:** Primer sequences with product size and accession number were analyzed via real-time PCR. Genes: glyceraldehyde-3-phosphate dehydrogenase (GAPDH), interlukin-1β (IL-1β), inter-lukin-2 (IL-2), interlukin-4 (IL-4), interlukin-6 (IL-6), interlukin-10 (IL-10), inter-lukin-12 (IL-12), and interferon-γ (IFN-γ).

Gene	Primer	Sequences	Product Size	Accession Number
			(Base Pairs)	
GAPDH	F	GTTCAACGGCACAGTCAAGGCAGAG	123	NM_001034034
	R	ACCACATACTCAGCACCAGCATCAC		
IL-1β	F	GCCTACGCACATGTCTTCCA	111	NM_174093
	R	TGCGTCACACAGAAACTCGTC		
IL-2	F	TGCTGGATTTACAGTTGCTT	111	XM_024976996
	R	TCAATTCTGTAGCGTTAACCT		
IL-4	F	ATCAAAACGCTGAACATCCTC	142	NM_173921
	R	TCCTGTAGATACGCCTAAGCTC		
IL-6	F	AGCTCTCATTAAGCGCATGG	168	NM_173923
	R	ATCGCCTGATTGAACCCAG		
IL-10	F	GGCCTGACATCAAGGAGCAC	103	NM_174088
	R	CTCTTGTTTTCGCAGGGCAGA		
IL-12	F	CCGCATTCCTACTTCTCCCT	178	XM_027545793
	R	ACACAGATGCCCATTCACT		
IFN-γ	F	TGATTCAAATTCCGGTGGAT	108	NM_174086
	R	TCTTCCGCTTTCTGAGGTT		

## Data Availability

The data presented in this study are available on request from the corresponding author.

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
