# Peer review of "Effects of Maternal Supplementation with Organic Trace Minerals including Zinc, Manganese, Copper, and Cobalt during the Late and Post-Partum Periods on the Health and Immune Status of Japanese Black Calves"

_animals, 2023, doi:10.3390/ani13233679_

Round 1

Reviewer 1 Report

Comments and Suggestions for Authors

Line 19-26: Beef or dairy calves? Please indicate in the summary.

Line 20-21: This sentence reads strangely. Calf health “was” prioritized is not correct. Calf health should always be prioritized.

Line 21: Suggest changing “post-gestation” to postpartum.

Line 25: What does “highly-producing” mean? Suggest changing for clarity.

Line 27-42: Beef or dairy calves? Please indicate in the abstract. Additional clarity is requested about experimental design. Did control cows have no trace minerals? How long were treatments administered for? Authors should rewrite the abstract to properly reflect materials and methods in order to make conclusions.

Line 29: How were cows fed, individually, group?

Line 59: “severe symptoms” is far too vague. Please be specific.

Line 62-65: Cattle are not frequently supplemented with organic trace minerals, sulfate forms are the most common. Organic trace minerals are hypothesized to have greater bioavailability, it is not proven repeatedly. Please revise.

Line 67: Higher liver concentrations of these trace minerals when? Throughout life? At birth? Please make clear.

Line 68: What is growth rate at weaning? Do the authors mean weaning body weight? This is unclear.

Line 70-75: Please revise to accurately reflect the literature cited. These two were completely different and separate studies, the text currently reads as though these are one study. Further, the two citations are reviews. Please be clear.

Line 82: Please change post-gestation to postpartum or lactation. Post gestation is ambiguous.

Line 92-95: Please provide this information in a table, along with diet composition.

Line 96-97: How were cows assigned to treatment? The authors have provided no indication of cow body weight, age, body condition score etc.

Line 100: How were cows fed? In a group? Individually? Were treatments top dressed? The authors have provided no indication of cow nutritional regimen. Were cows limit fed? If not, was dry matter intake evaluated? Proper explanation of experimental design is warranted before further review.

The authors have provided limited explanation of experimental design, hence hindering proper statistical analysis. Further, authors have not provided sufficient explanation of statistical analysis, which is warranted for publication in this journal rendering the manuscript unsuitable for publication.

Comments on the Quality of English Language

Moderate editing is required.

Author Response

Answers to Reviewer 1:

We appreciate the time and effort that you dedicated to providing feedback on our manuscript and are grateful for the insightful comments on and valuable improvements to our paper. Our answers to your comments are listed below following each specific question/comment. In the manuscript, the revised portion of this time are shown in red color.

  • Line 19-26: Beef or dairy calves? Please indicate in the summary.

Response: Thank you for your comment. We added beef in line 19 and 24.

  • Line 20-21: This sentence reads strangely. Calf health “was” prioritized is not correct. Calf health should always be prioritized.

Response: We have changed the sentence and adjusted it in line 20.

  • Line 21: Suggest changing “post-gestation” to postpartum.

Response: We have changed from post-gestation to postpartum in line 21.

  • Line 25: What does “highly-producing” mean? Suggest changing for clarity.

Response: We changed the sentence and added in line 25.

  • Line 27-42: Beef or dairy calves? Please indicate in the abstract. Additional clarity is requested about experimental design. Did control cows have no trace minerals? How long were treatments administered for? Authors should rewrite the abstract to properly reflect materials and methods in order to make conclusions.

Response: Thank you for your comment. We added beef in line 27. And we added in line 29 the duration of treatment. The control group's diet was free of supplemental trace mineral preparations of Zn, Mn, Cu, and Co. In addition, we added experimental design in Figure 1.

  • Line 29: How were cows fed, individually, group?

Response: Cows were fed individually not in groups as described in line 102.

  • Line 62-65: Cattle are not frequently supplemented with organic trace minerals, sulfate forms are the most common. Organic trace minerals are hypothesized to have greater bioavailability, it is not proven repeatedly. Please revise.

Response: Thank you for your comment. We changed the sentence in line 63.

  • Line 67: Higher liver concentrations of these trace minerals when? Throughout life? At birth? Please make clear.

Response: We added at birth in the text in line 69.

  • Line 68: What is growth rate at weaning? Do the authors mean weaning body weight? This is unclear.

Response: We clarified in the text the body weight of newborns in line 69.

  • Line 70-75: Please revise to accurately reflect the literature cited. These two were completely different and separate studies, the text currently reads as though these are one study. Further, the two citations are reviews. Please be clear.

Response: We revised the text and changed in line 72.

  • Line 82: Please change post-gestation to postpartum or lactation. Post-gestation is ambiguous.

Response: Thank you for your comment. We have changed in text (line 83).

  • Line 92-95: Please provide this information in a table, along with diet composition.

Response: We provided the nutrient formula with diet composition in Table S1 as supplementary data in the text (line 95).

  • Line 96-97: How were cows assigned to treatment? The authors have provided no indication of cow body weight, age, body condition score etc.

Response: The pregnant JB cows were used in August 2021, and they were allocated to the control and the treated group alternately in the order of their scheduled date of calving. The cow’s weight did not vary in appearance and were generally around 500 kg, and the body condition score is around 3 from 5 added in lines 90-91. The average age at calving period:  control group (3.8±1.1 years old) and treated group (3.7±1.1 years old) also added in line 98.

  • Line 100: How were cows fed? In a group? Individually? Were treatments top dressed? The authors have provided no indication of cow nutritional regimen. Were cows limit fed? If not, was dry matter intake evaluated? Proper explanation of experimental design is warranted before further review.

Response: When feeding with supplement starts the cows will come to dome barn A (stanchion and free barn) 45 days before calving is scheduled. They will be in the stanchion barn at feeding time and cows individually receive their assigned supplement. Between 30 and 15 days before the scheduled calving date, the cow and calf stayed in a private stall, so they ate completely individually. The supplement was mixed with the feed. Not free feeding, cows were fed twice a day (morning and evening). The nutritional requirements are based on 8th edition of the Japanese beef cattle feeding standard (NRC, 2000), we added the nutritional regimen in Table S1 (supplementary data).

  • The authors have provided limited explanation of experimental design, hence hindering proper statistical analysis. Further, authors have not provided sufficient explanation of statistical analysis, which is warranted for publication in this journal rendering the manuscript unsuitable for publication.

Response: We added provided the experimental design as shown in Figure 1, and the cows management as supplementary data. Statistical analysis of differences between the control and treated groups was performed using the Mann Whitney U test. And results are expressed as the mean ± SD. We used nonparametric test (Log-rank test) was used to determine whether the difference in the incidence rate between the experimental groups. We added in line 167-169.

Reviewer 2 Report

Comments and Suggestions for Authors

This manuscript describes relevant points clearly and tackles the knowledge gap of beef cow late gestation organic mineral supplementation effect on the calves adequately. The benefits are shown nicely. There could be done some clarification on the materials and methods section on the management practices.

Author Response

Answers to Reviewer 2:

We appreciate the time and effort that you dedicated to providing feedback on our manuscript and are grateful for the insightful comments on and valuable improvements to our paper. Our answers to your comments are listed below following each specific question/comment. In the manuscript, the revised portion of this time are shown in red color.

  • Line 88: 19 healthy JB pregnant beef cows was the pedigree (dam sire eg.) similar for all the experiment cows? See below the effect of pedigree on the immunological function.

Response: Thank you for your comment, based on the information from the farm, in our study we used different pedigrees. There are six different sires in the control group and three different sires in the treated group. Previous studies reported that there are some factors affecting on the immunological functions such as age, period, strain, and malnutrition. Ohtsuka et al, reported that there is difference in immunological functions between JB cattle and Holstein cattle. For immunological status in JB young calves, nutrition and health condition of caws are very important factors.

  1. Ohtsuka H, Ono M, Saruyama Y, Mukai M, Kohiruimaki M, Kawamura S. 2011. Comparison of the peripheral blood leukocyte population between Japanese Black and Hol- stein calves. Animal Science Journal 82, 93–98.
  2. Ohtsuka H, Kobayashi H, Kinouchi K, Kiyono M, Maeda Y. Comparison of cytokine mRNA expression in peripheral CD4(+) , CD8(+) and γδ T cells between healthy Holstein and Japanese Black calves. Anim Sci J. 2014 May;85(5):575-80. doi: 10.1111/asj.12175. Epub 2014 Feb 8. PMID: 24506816.
  • Line 88: Management practices during last period of the pregnancy and after calving? Where the cows housed? Confined? Free stall? How much area/cow? Tie stall? Grazed?

Response: In our study, the cows were housed in dome barn A (stanchion, free barn) 45 days before the expected calving date. After that, from 30-15 days before calving, the cows were housed in private rooms. After calving, the mother and calf are moved to dome barn B (stanchion of free barn). We provided the cows management practices during the last trimester of pregnancy and after calving as supplementary data.

  • Line 89: The same AI bull for all cows? Is this the sire of all the calves? There is indication in the literature that there are heritable differences in immunological function in the offspring. If there are multiple sires used this might have an effect on the outcome of the study especially when the number of the animals was rather low- How many days postpartum was the AI done? Via natural heat cycle or synchronized?

Response: Thank you for the comment, some cows were pregnant from AI for the same sire, and others from different sires. The average number of days since calving for AI is as follows, Control group: 76 days and treated group: 78 days. The artificial insemination was done in some cows were in normal estrous and others were in synchronized estrous (Control group: 5 normal estrous and 4 synchronized estrous, treated group: 8 normal estrous and 2 synchronized estrous).

  • Line 92: I wonder on what bases the nutrient requirements were decided. No description how the dry matter intake of the beef cows was calculated. Liveweight based? Were the beef cows weighed? Were the beef cows condition scored? Was the condition score same for the beef cows in the feeding groups?

Response: Nutritional requirements are based on the 8th edition of the Japanese beef cattle feeding standard (NRC, 2000). In addition, the dry matter of beef cows was calculated based on the 8th edition of the Japanese beef cattle as shown in Table S1 in line 95 (supplementary data). The live weight of the mother cow was not measured, there is no variation in the weight of the cows from the appearance, but they are generally around 500kg, and the body condition score doesn’t differ among the herds is approximately 3 from 5.

  • Line 100: How the cows were given the supplement? Free choice? In the TMR? After calving did the calves have access to the trace mineral supplement? Or the TMR? If TMR how the TMR was composed so the small amounts of supplement is evenly distributed in the fed TMR?

Response: The supplements were mixed with the feed. Since the animals are in stanchions and private rooms, farm staff can ensure that the animals are fed evenly and that they have eaten their entire meal. After calving, calves do not receive trace minerals because they do not reach the feed tank.

  • Chest circumference, respiratory and digestive disease incidence, and treatment
    Line 171-172: What respiratory/digestive diseases the calves were treated for? Veterinary diagnosis? What kind of indications of ill health was the treatment based on? Can not find anything on the diagnosis or on the treatment in the paper.

Response: The staff found a calf in poor condition (no milk, no energy, fever, etc.) and the veterinarian diagnosed and treated it.

  • Zn concentration
    Line 186: Did the calves have access to the trace mineral supplement after birth? TMR?

Response: on day 1, calves in the treated group had a higher serum Zn concentration. These findings indicate that maternal colostrum contains trace minerals that are transferred to neonatal calves during the first day of life. However, calves do not receive trace minerals because they do not reach the feed trough after birth. We added the experimental design in Figure 1.